# A Proposed Hierarchy of Deep Learning Tasks

## Abstract

As the pace of deep learning innovation accelerates, it becomes increasingly important to organize the space of problems by relative difficultly. Looking to other fields for inspiration, we see analogies to the Chomsky Hierarchy in computational linguistics and time and space complexity in theoretical computer science.

As a complement to prior theoretical work on the data and computational requirements of learning, this paper presents an empirical approach. We introduce a methodology for measuring validation error scaling with data and model size and test tasks in natural language, vision, and speech domains. We find that power-law validation error scaling exists across a breadth of factors and that model size scales sublinearly with data size, suggesting that simple learning theoretic models offer insights into the scaling behavior of realistic deep learning settings, and providing a new perspective on how to organize the space of problems.

We measure the power-law exponent—the "steepness" of the learning curve—and propose using this metric to sort problems by degree of difficulty. There is no data like more data, but some tasks are more effective at taking advantage of more data. Those that are more effective are easier on the proposed scale.

Using this approach, we can observe that studied tasks in speech and vision domains scale faster than those in the natural language domain, offering insight into the observation that progress in these areas has proceeded more rapidly than in natural language.

## 1 Introduction

There are so many new exciting deep learning results published every day that it is hard to see the big picture. Many of these papers report exciting performance on a particular solution to a particular task, but it is hard to know how well those results are likely to generalize beyond the particulars in a particular paper. In undergraduate classes on Algorithms, we are taught how to reduce one problem to another, so we can make claims about time and space complexity that generalize across a wide range of problems.

It would be much easier to make sense of the deep learning literature if we could find ways to generalize more effectively across problems. What can nets do, and what can't they do? Which tasks are relatively easy and which are relatively hard? Which problems require more resources?

Is it possible to come up with an organization of the deep learning literature that is somewhat similar to the Chomsky Hierarchy, Chomsky (1957), in Table 1? The Chomsky Hierarchy makes it clear what can (and cannot) be done within broad classes of computational resources. Finite state machines can do many things, but they can't solve the Halting Problem (because nothing can solve the Halting Problem). Finite state machines also can't solve problems that take more than constant space and linear time such as sorting large vectors, multiplying large matrices, and more.

We would like to come up with statements about deep nets that generalize across topics. In particular, it might be useful to view certain exciting deep nets like convolutional neural nets as (large) finite state machines, and therefore, some of the statements mentioned above may apply to these nets. Obviously, there are well-known ways to go beyond finite-state, such as (Sun et al. (2017b)), but doing so is not necessarily a good thing (because doing so tends to lead to increases in computational costs). Much of the excitement in neural nets is focused on recent progress in speech and vision, where it may not be necessary (and even desirable) to go beyond finite state.

Table 1: Chomsky Hierarchy

| Automata | Languages | Time | Space |
|---|---|---|---|
| Finite State (FSA) | Regular | $O(n)$ | $O(1)$ |
| Push Down (PDA) | Context Free (CF) | Matrix Multiply | Matrix Multiply |
| Linear Bounded | Context Sensitive (CS) | Worse | Worse |
| Turing Machines | Recursively Enumerable | Beyond Worse | |

That said, it may not be all that helpful to focus too much on time and space, since training data appears to be more of a limiting factor on progress than time and space complexity. This paper will propose an organization of the deep nets literature based on training data, as opposed to time and space. The ranking of problems is intended to make it easier to generalize across tasks. If we have a new problem, can we compare it to a bunch of known problems in a meaningful way? Can this ranking help us estimate how easy or hard the new problem is? How do we estimate requirements on computational resources for a new problem?

This paper will suggest a particular proposal for ranking deep net problems based on learning curves. In general, there is no data like more data, but we find that some deep net tasks are more effective than others in taking advantage of more data. We suggest fitting learning curves to a power law, and then sorting tasks by empirical estimates of exponents. That is, we assume that loss, $\mathcal{E}(m)$, can be modeled as a function of training set size, $m$, with: $\mathcal{E}(m) = \alpha m^{\beta_g}$. We propose sorting problems by the exponent: $\beta_g$.

Dimensionless quantities (like $\beta_g$) are convenient for making comparisons across different problems. In general, different problems use different units. For example, in computational linguistics, it makes sense to measure the size of the training set in terms of words and/or characters, and to measure loss in terms of perplexity, but these metrics generally do not generalize well to other problems such as speech and vision. Since $\beta_g$ is dimensionless, it is more suitable for comparisons across problems than alternatives such as $m$, $\mathcal{E}(m)$, $\alpha$, where the units tend to vary from one problem to another.

$\beta_g$ can depend on a number of factors including task and metric. We not only see important differences in $\beta_g$ across tasks, but we also see differences across metrics. As we will see in Section 4.1, $\beta_g$ for top-1 classification can be different from $\beta_g$ for top-5 classification, even on the same task.

It isn't easy to measure $\beta_g$ over lots of solutions to lots of problems. It took about 50 GPU years to estimate the $\beta_g$s in this paper. We not only considered a range of different problems, but also a range of different models (with more or less capacity), optimizers, regularizers and loss functions. We will show examples where different solutions improve the constants, $\alpha$, but we are more interested in improvements in $\beta_g$.

An interesting question for future work is why $\beta_g$ is different in different cases. Our estimates of $\beta_g$ are similar to estimates of cross-entropy. It is common practice with cross-entropy to try a number of different models, and return the best of those attempts. Cross-entropy is bounded by the true entropy of the task. There is a gap between the two because there may be a better solution than the ones that we considered. So too, with estimates of $\beta_g$, it is hard to separate fundamental properties of the problems from concerns that there may be a better solution than the ones we considered.

The main point of this paper is to start a discussion in search of a big picture that will eventually lead to agreement on how to generalize beyond the particulars in a particular deep net paper. It would be nice if our particular proposal is adopted, but it is more important to us that the field agree on a satisfactory solution than that they adopt our particular proposal.

This paper is organized as follows: We begin with a review of related work measuring relationships between data set size, model size, and error in Section 2. Section 3 describes our methodology, and Section 4 shows learning curve results from the five tasks. Section 5 applies our proposed organization to these tasks.

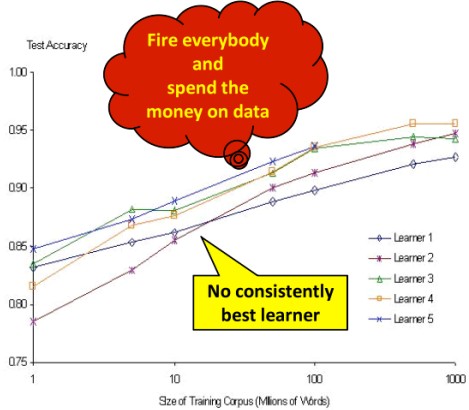 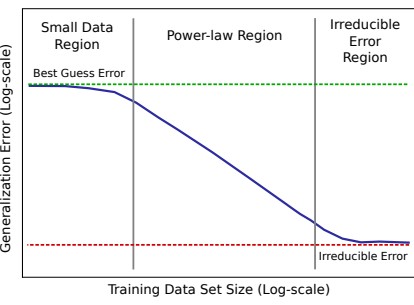

Figure 1: There is no data like more data. More training data improves performance (left) and reduces loss (right). The left panel is borrowed with permission from Banko and Brill (2001). Brill has suggested firing everyone to fund data collection (personal communication), perhaps in jest, though that comment is not in the published paper.

## 2 RELATED WORK

**Empirical Studies:** Figure 1 compares a well known result from (Banko and Brill (2001)) (left) with a cartoon sketch of a power law (right). Banko and Brill (2001) test a language modeling problem (confusion set disambiguation) trained using subsets of a billion-word corpus of text. Their results show that there is no data like more data. Performance on this disambiguation task improves with the size of the training set, roughly consistent with a power law. They note that some learning methods are slightly better at one operating point, and others are slight better at another operating point. But those small differences are dominated by a more important trend. A rising tide lifts all boats; more data improves performance for all methods. More data is more important than relatively small differences between methods.

Power-law-like learning curves have been reported by a number of researchers for a number of applications. Jelinek refers to Banko and Brill (2001) and Lamel et al. (2002) in his speech accepting the Zampolli prize at LREC-2004.[1] Amodei et al. (2016) show an improvement in WER (word error rate) for a Deep Speech 2 model with more and more training data; Sun et al. (2017a) report similar improvements for object detection and semantic segmentation tasks.

The cartoon on the right of Figure 1 shows a similar power-law pattern for loss (as opposed to accuracy). More training data improves performance (left) and reduces loss (right). The cartoon on the right splits the learning curve into three regions. We are most interested in the middle region where the power-law assumption is more appropriate. The power-law assumption is less appropriate at the extremes where there is too much (or too little) training data. It is common in practice to have too little training data, but it is harder to find realistic examples with too much training data. In this study, we have been able to construct toy problems where the power law assumption breaks down with extremely large training set sizes, but we have failed to find that extreme case in real world examples in the wild.

**Theoretical Analysis:** There is a considerable literature deriving theoretical bounds on the generalization gap between expected and empirical error. At least five lines of work derive bounds that follow power laws on the number of training samples:

1. Hypothesis space bounds in PAC learning (Haussler (1988)) using Vapnik–Chervonenkis (VC) dimension (Ehrenfeucht et al. (1989); Blumer et al. (1989); Haussler et al. (1996)),

2. Rademacher complexity in (Bartlett and Mendelson (2002) and Mohri et al. (2012)),

3. the *stability* approach in (Bousquet and Elisseeff (2002)) considers the dependence of the learned model on the training dataset, as well as how much individual changes to the training dataset can affect the learned model,

---

[1] http://www.lrec-conf.org/lrec2004/doc/jelinek.pdf

4. the *robustness* approach in (Xu and Mannor (2012)) measures how much the error can vary with the input space,

5. (Amari et al. (1992) and Amari (1993)) derive bounds using statistical mechanics.

Kawaguchi et al. (2017) provides a clear survey of much of this prior work.

We follow the tradition in much of this literature of introducing a power law assumption. In spite of the complex nature of real tasks, the reliable observation of power-law learning curves in our experiments suggests that deep learning exhibits the same scaling phenomena as those that are simple to analyze theoretically.

**Model Capacity Required to Fit Data**: This work will assume that both loss and model capacity obey power laws. Prior studies propose various measures of model capacity based on a model's organization and parameterization, and these measures hint at the model size required to fit a training set. Vapnik and Chervonenkis defined the VC dimension of a model as the cardinality of the largest set of data points that a model can shatter (Vapnik (1998)). Follow-on work uses data complexity measures to estimate the structure of model families that might fit the data (Bartlett and Mendelson (2002)). Recent work explores applying these concepts to deep nets (Harvey et al. (2017); Dziugaite and Roy (2017); Collins et al. (2017)).

Prior work to empirically estimate model scaling with training set size is very sparse. In our experiments, we find power-laws produce good fits, i.e. $s(m) \propto \alpha m^{\beta_p}$, where $s(m)$ is the required model size to fit a training set of size $m$, and $\beta_p \in [0.5, 1]$. When applying this approach to the data in Banko and Brill (2001), we find that the Winnow and memory-based models grow with the same power-law exponent to larger data sets, $\beta_p \approx 0.72$.

## 3 METHODOLOGY

Ideally we are interested in learning curves that depict the relationship between the expected error of models in the wild and the amount of training data. However, expected error in a real setting cannot be measured. So we follow the common practice of approximating it with empirical error computed on a held-out validation set (this choice explained further in Appendix B), and plot this instead. We divide a training set into independent shards of different sizes, perform a hyper-parameter grid search over optimizer parameters and model sizes to find the best model for each shard, and plot the validation error of that model.

We ensure that the training set is randomly shuffled so that shards will have a similar data distribution. We use the same validation set for all models, and size it empirically such that measurements have low variance. Depending the task, error metrics include cross-entropy, $L^p$ norms, and classification error (see Appendix A for more details). We search over different model sizes by varying the hidden dimension, but do not make other changes to the architecture except where noted. We use early stopping for regularization in all cases.

## 4 LEARNING CURVE RESULTS

We present empirical results characterizing learning curves for machine translation, language modeling, image classification, and speech recognition tasks. We focus our discussion on image recognition and word language modeling to highlight example tasks that learn relatively fast and slow with data. The other tasks and experiments are described in more detail in Appendix C.

### 4.1 IMAGE CLASSIFICATION

We begin with image classification, a well-studied task that aims to identify objects in high-dimensional image data. Image classification is used in applications such as object recognition, image captioning, and tagging video content. Image classification shows clear power-law learning curves and model size scaling relationships. We also show that accuracy plateaus near random guessing on very small training sets.

We test ResNets (He et al. (2016); Wu et al. (2016)), which are popular architectures for ImageNet classification (Russakovsky et al. (2015)). ResNets are deep networks built from blocks containing

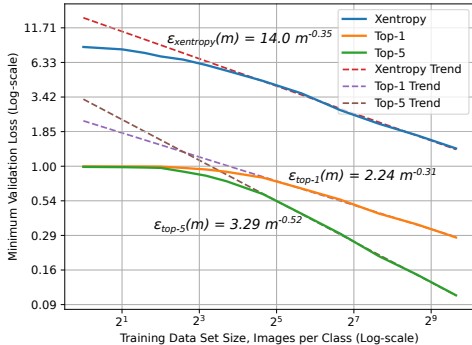 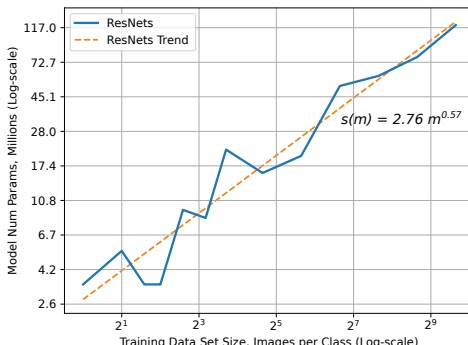

Figure 2: Learning curve (left) and model size (right) results and trends for ResNet image classification. Note the transition from the small data regime dominated by best guessing to the power-law scaling around 32,000 training images.

convolutions, nonlinearities, and pooling layers. They have residual connections from the inputs to outputs of most blocks that permit the network to bypass layers. We train and validate ResNets on various shard sizes of ImageNet, ranging from 1 image per class (0.08% of images) up to 800 images per class (62%). ImageNet has 1,000 different object classes as outputs.

We start with five known variants of ResNets with depths 18, 34, 50, 101, and 152 layers. We first scale the model sizes by changing the number of layers ranging from 10 to 200. To provide even finer-grained model size control, we also change the number of convolution filters using a scaling factor. We scale filter counts proportionally across all convolution blocks with scaling factors 0.0625 to 1.5. We test models with parameter counts ranging from 89K to 121M. We use a Nesterov Momentum optimizer targeting classification cross-entropy loss. We remove weight regularization.

Figure 2 (left) shows that various loss calculations follow the power-law learning curves. We report average validation cross-entropy, top-1, and top-5 classification errors. For small training sets—less than roughly 25 images per class—these error metrics are roughly equal to the model random guessing (i.e., greater than $-log(1/1,000) \approx 6.9$ for cross-entropy, and near $1 - (1/1,000) = 99.9\%$ classification error for top-1 and top-5). Models are unable to extract enough information from these small training sets to make many accurate classifications on the validation set. This is an example of the "small data region" explained in Section 1.

As long as the training set is large enough, we observe that generalization improves on a power-law, but the power-law exponent is different for each of the reported metrics, as mentioned in Section 1. The top-1 classification error exponent is $\beta_g = -0.309$, the exponent for top-5 classification error is $\beta_g = -0.522$, and the validation cross-entropy exponent is $\beta_g = -0.35$. Figure 2 (right) shows that model size growth is well fit by a power-law. The best-fit ResNet models grow following a sublinear curve with exponent $\beta_p = 0.573$.

## 4.2 LANGUAGE MODELING

Language models (LMs) aim to predict probability distributions for the next character, word, or other textual grams conditioned on a previous sequence of input text. LMs are very important model features for domains such as speech recognition and machine translation, helping to identify most probable sequences of grams. LMs have low-dimensional input and output spaces, and can be trained with very large labeled sets.

LM learning curves and model size scaling relationships are very robust, and the power-law exponents tend to be small ($\beta_g \in [-0.09, -0.06]$). These small exponents indicate that current language models will require significantly more data to significantly improve accuracy. The models that give the best validation error grow sublinearly in the training set size ($\beta_p \approx 0.7$).

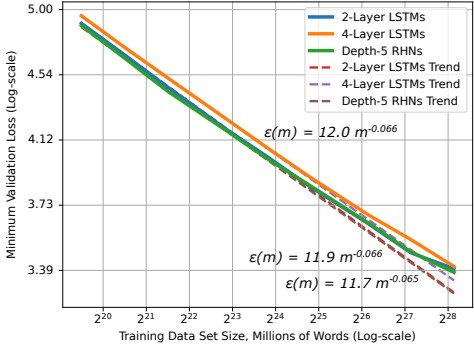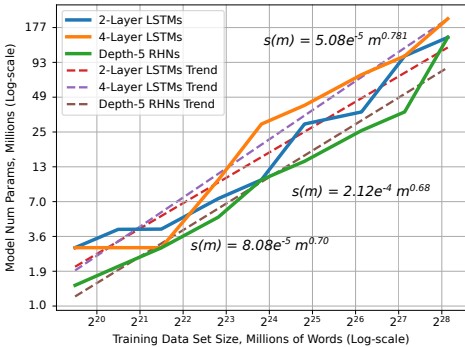

Figure 3: Learning curve (left) and model size (right) results and trends for word language models. Note that all models exhibit the same $\beta_g$ regardless of depth or cell type (LSTM vs RHN).

### 4.2.1 WORD LANGUAGE MODELS

We train LSTM-based word LMs as described in Jozefowicz et al. (2016) with some small changes. To reduce the computational requirements of the models, we restrict the vocabulary to the top 10,000 most frequent words in the Billion Word Dataset (Chelba et al. (2013)). The networks are 2- or 4-layer LSTMs with the same number of hidden weights in each layer, and we scale the number of layer weights to modulate the model size and find the best fit model for each training shard size. We also compare LSTMs against Recurrent Highway Networks (RHNs) described in Zilly et al. (2017). Specifically, we train single-layer, depth 5 RHNs to see if the different network organizations show different generalization trends. We use a stochastic gradient descent optimizer (SGD) with per-sequence cross-entropy loss, and we report per-predicted-word average cross-entropy loss. We do not use dropout. We train the models on shards ranging from 0.1% up to 40% of the Billion Word Dataset.

Figure 3 shows the learning curve and model size results for LSTM and RHN word language models. First, the loss scaling relationships are smooth power-law functions of the data set size with almost exactly the same exponents: $\beta_g = -0.0656 \pm 1\%$. Larger models are more computationally expensive, and we have more difficulty optimizing to fit the larger training sets given our compute resources. The best tuned models settle at or just above the power-law trend, and we believe that further hyperparameter search is likely to yield a model on the trend.

Strikingly, although these model architectures differ appreciably, they all show the same learning curve profile characterized by the power-law exponent. Increasing the LSTMs depth from 2 to 4 layers decreases the networks' accuracy by about $1.5\%$, but both model architectures see the same relative loss improvement as we increase training set size. RHNs have significantly different recurrence structure than LSTMs, but show nearly identical learning curves.

Model size results show that best-fit models grow sublinearly in the training shard size. Specifically, the best-fit 2-layer LSTM and depth-5 RHNs model sizes grow roughly with $\beta_p = 0.69$. The 4-layer LSTMs show slightly worse scaling with $\beta_p = 0.89$, suggesting they make less effective use of extra parameters on larger data sets. Despite the model size scaling differences, for a given model architecture, we can accurately predict the model size that will best fit increasingly larger data sets.

These results underscore the importance of continued data and computational scaling for the natural language domain.

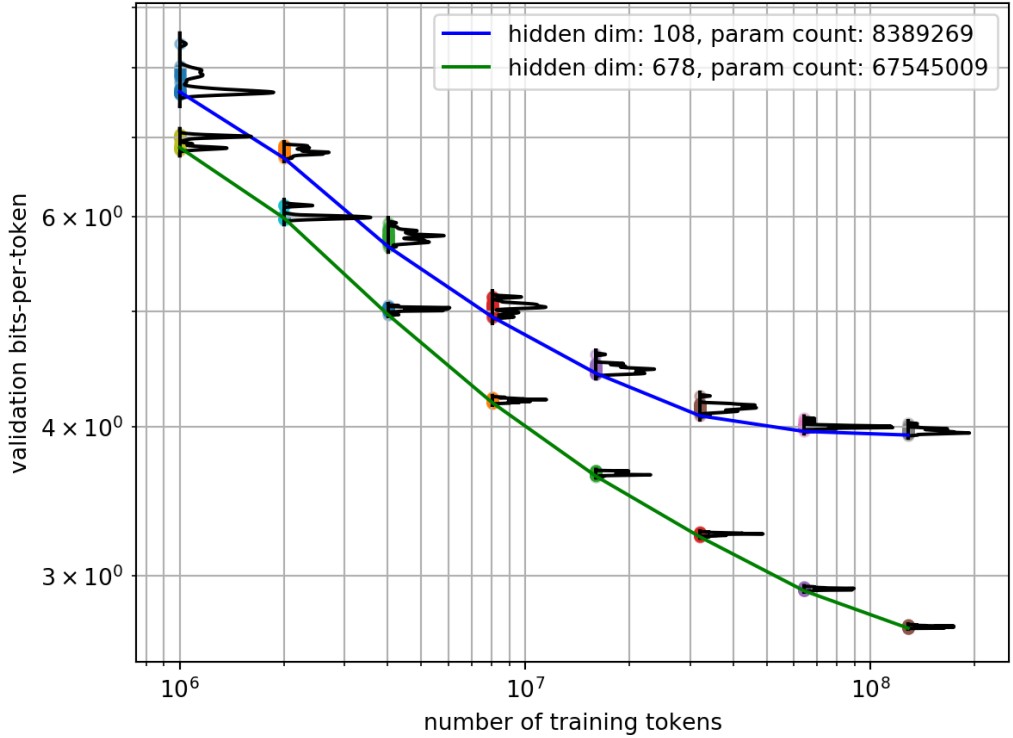

Figure 4: Violin plots of learning curves for neural machine translation. For each training data shard size, the violin plot shows the distribution of validation errors for 30 repetitions of the experiment. Note that overall the variance is small relative to the reduction in validation error, and that the variance is reduced with more training data and also with larger models.

## 4.3 SENSITIVITY OF RESULTS

Given that we present empirical measurements of scaling, it would be valuable to estimate the basic statistics of our scaling model parameters $(\alpha_g, \beta_g, \gamma_g, \alpha_p, \beta_p)$. However, our experiments collectively used about 50 years of GPU time, so generating each sample is extremely expensive. For example, computing confidence intervals for each parameter using 30 samples would require 1500 GPU years of compute time, and we don't have that many resources. In this section, we perform additional experiments requiring about two months of training time to study the repeatably of our results for one domain.

Figure 4 uses a violin plot to show the distribution of validation errors across 30 repetitions of the machine translation experiment with 8M and 67M parameters with different seeds for weight initialization and shard creation, for each training data shard size. We find that overall the variance is small relative to the reduction in validation error for each shard, and that the variance is reduced with more training data and also with larger models (similar to the observations in Choromanska et al. (2014)). This suggests that our results are repeatable and will become even more stable for larger datasets.

### 4.4 SUMMARY OF RESULTS

Our results are summarized in Table 2, illustrating how $\beta_g$ (and $\beta_p$) can be used to make comparisons across a wide range of diverse tasks. Less is more. Smaller $\beta_g$s are better. In general, there is no data like more data, but some tasks are more effective than others in taking advantage of more data. Note that speech and vision have better $\beta_g$s than other tasks mentioned in Table 2. It is probably not an accident that the field is relatively excited about the tasks with better $\beta_g$s.

The tasks with better $\beta_g$s also tend to have better $\beta_p$s. That is, there appears to be a relationship between parameter requirements ($\beta_p$) and generalization/effectiveness in taking advantage of more data ($\beta_g$). $\beta_p$ is particularly worrisome for Language Modeling on characters; ideally, nets ought to be learning models that are considerably smaller than the training set ($\beta_p \ll 1$), but with $\beta_p = 0.89$, there are almost as many parameters as training samples.

| Task | Learning Curve $\beta_g$ | Model Size $\beta_p$ |
|------|--------------------------|----------------------|
| Language Modeling (Words) | −0.066 | 0.68 |
| Language Modeling (Characters) | −0.092 | 0.89 |
| Machine Translation | −0.128 | 0.68 |
| Speech Recognition | −0.291 | 0.54 |
| Image Classification | −0.309 | 0.57 |

Table 2: The proposed hierarchy: a ranking of deep learning tasks sorted by $\beta_g$. In general, there is no data like more data, but some tasks are more effective than others in taking advantage of more data. Note that speech & vision have better $\beta_g$s than language modeling. This may help explain why there is relatively more excitement about deep nets in speech and vision (vs. language modeling).

## 5 DISCUSSION

It is interesting that the order produced by sorting applications by $\beta_g$ is aligned with the observation that progress in computer vision and speech recognition has proceeded relatively quickly. We note that both of these tasks use relatively high dimensional input data. Studied speech and vision tasks learn faster with additional training data. Clearly, there appears to be scope for all of these tasks to further improve accuracy by training on more data, but these results suggest that the computational and data requirements for language modeling in particular will be more challenging.

We are left with the open question of whether or not it is possible for learning algorithms to improve $\beta_g$ or $\beta_p$. In our results we noted that $\beta_g$ appeared stable across changes in neural network architecture, regularization method, and optimization algorithm. It changed between tasks such as language modeling vs. image classification and between metrics such as top-1 vs. top-5 classification error. We hypothesize that it is bounded by properties of the task setup and data distribution, and challenge future work to precisely determine the factors that affect it. Regarding $\beta_p$, simply memorizing the dataset would yield $\beta_p = 1.0$, and it is interesting to note that all studied algorithms improve on this. There may be scope in future work to further improve on it beyond the models that we have studied.

There is also clearly scope for a more detailed organization of tasks within important domains. We can already see that machine translation has a smaller $\beta_g$ than language modeling in spite of both working with low dimensional natural language data. We expect future work to use this methodology to guide efforts on the most valuable tasks that can be improved the most quickly.

## 6 CONCLUSION

The Chomsky Hierarchy in computational linguistics and time and space complexity in theoretical computer science have been instrumental in guiding work on open problems. We have measured the power-law exponent ($\beta_g$) —the "steepness" of the learning curve—and propose using this metric to rank problems by degree of difficulty. There is no data like more data, but some tasks are more effective at taking advantage of more data. We find that the problems that the field is most excited about in speech and vision have better $\beta_g$s than problems that the field is less excited about (such as language modeling). When applied to an even wider set of problems, we hope that such a hierarchy can serve as a guide to the data and computational requirements of open problems.

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

# A  DETAIL ON TESTED MACHINE LEARNING TASKS

To encourage further investigation, this section reports precise definitions of input and output spaces, optimized and reported loss functions for each machine learning domain, and other information that may be relevant to predicting learning curves and model size scaling. Additionally, to show the breadth of our testing, Table 3 summarizes the different domains, model architecture features, optimization and loss functions we tested.

Table 3: Breadth of domains, model features, optimizers, loss functions tested

| Domain | Model | Model Features | Optimizer | Loss Function | Exponent ($\beta_g$) |
|---|---|---|---|---|---|
| Machine Translation | LSTM | Encoder-decoder with attention, with and without dropout | Adam | Token Error | $-0.128$ |
| Word LMs | LSTM | GEMMs, $\sigma + tanh$ non-linearities | SGD | Xentropy | $-0.066$ |
| | RHN | GEMMs, $\sigma + tanh$ non-linearities | SGD | Xentropy | $-0.070$ |
| Char LMs | RHN | GEMMs, $\sigma + tanh$ non-linearities | SGD, Adam | Xentropy | $-0.094$ |
| Image Classification | ResNet | Feed-forward, CONV blocks, pooling and skip connections | Nesterov Momentum | Classify Error | $-0.309$ |
| | | | | X-entropy | $-0.350$ |
| Speech Recognition | DS2 | Bi-LSTM, CTC loss | Adam | CER | $-0.299$ |
| | Attention | Bi-LSTM, CONVs, attention layer | Adam | CER | $-0.296$ |

## A.1  NEURAL MACHINE TRANSLATION

Given input and output vocabularies, $V_S$ and $V_T$, NMT models learn a mapping $D_S \to D_T$ where $D. = V.^*$ (Kleene star). In this work, we use a word-piece vocabulary shared between the source and target languages. After applying pre-processing methods[2] adopted in many recent models, there are 36545 sub-word tokens. We include UNK and PAD tokens for unknown words and minibatch padding for the source domain (German, $|V_S| = 36547$); for the target domain (English), UNK, PAD, SOS (start-of-sequence), and EOS (end-of-sequence) are included ($|V_T| = 36549$). The German and English sentences in newstest2016 were on average 27 and 25 tokens long with the longest sequences having 101 and 94 tokens respectively.

During training, we minimize cross entropy loss (i.e. the conditional probability of the target sentence given the source sentence). We report per-token error rate and bits-per-token. Because our reported metrics are per-token measure of the target language, the dataset size is given by the number of English tokens in the training set.

## A.2  LANGUAGE MODELING

### A.2.1  WORD LANGUAGE MODELS

During training for world language models, we unroll sequences out to length 80 for backpropagation. We also use continuous minibatching: At end of one sentence in the data set, we concatenate an end-of-sentence designator, followed by the next sentence from the data set.

Let $C$ be the language's vocabulary. Then, $|C| = 10,004$ after we include special symbols like the unknown token. The input space is $I = \bigcup C^i$ where $i$ is the number of words previously seen in a sequence. We use continuous minibatching, so the effective history length, $i$, can be very long. The output space is $O = C$.

Rather than perplexity, we use normalized cross-entropy loss: $-\frac{1}{N} \sum_i ln \ p_{w_i}$, where $p_{w_i}$ is the model's predicted probability of seeing the $i$th token. $N$ is either the number of sequences in a batch for training optimization or $N$ is the number of predicted words in the validation set.

---

[2] clean-up and byte pair encoding uses Tensorflow NMT WMT scripts

### A.2.2 CHARACTER LANGUAGE MODELS

For character language models, we unroll sequences out to length 150 characters. Unlike word language models, we use non-continuous minibatching, so some sequences end at an end-of-sentence token. Sequences longer than 150 characters are truncated.

Let $C$ be the language's vocabulary of alphanumeric characters and symbols. Then, $|C| = 98$ after we include special symbols like the end-of-sentence token. Similar to word language models, the input space is $I = \bigcup C^i$ where $i$ is the number of characters previously seen in a sequence. Since we use non-continuous minibatching, so the effective history length, $i$, is at most 150. The output space is $O = C$.

Similar to word language models, we use normalized cross-entropy loss: $-\frac{1}{N} \sum_i ln\, p_{w_i}$, where $p_{w_i}$ is the model's predicted probability of seeing the $i$th token. $N$ is either the number of sequences in a batch for training optimization or $N$ is the number of predicted characters in the validation set.

## A.3 IMAGE CLASSIFICATION

ImageNet images were initially scaled proportionally so that the shortest dimension of the image is 256 pixels. During training, these images are cropped to 224x224 as input to the CNN. Input images are 224x224 pixels by 3 color channels of 8 bits each. Thus, the total input space size is $|I| = 224 * 224 * 3 * 256 \approx 38.5M$. The output space is 1,000 different object classes that might be contained in the image. For training, we also augment the dataset by modifying the brightness, contrast, saturation, and lighting. In addition, we also flip the image horizontally. [3]

We optimize for classification cross-entropy loss on each training image, and we report average validation cross-entropy, top-1, and top-5 classification error. Each loss calculation still follows the power-law. However, we note that top-k classification error ($k > 1$) is not a distance metric; It uses set containment, which is not symmetric. Alternatively, it is a product of distance metrics, which is not necessarily a distance metric.

## A.4 SPEECH RECOGNITION

The audio input to speech recognition models can be represented as the sequence $x = (x_1, .., x_t)$ of length $t$. Each $x_i$ is an audio spectrogram over a small time window. Each predicted output is a character, encoded as a one-hot vector, $y_i$, representing the most probable text token at sequence step $i$. Output sequences are of the form $y = (y_1, ..., y_u)$. Models predict the conditional distribution $p(y|x)$ using an encoder-decoder form. Thus, $p(y|x) = \text{Decode}(\text{Encode}(x), y)$.

### A.4.1 DEEP SPEECH 2

In DS2 model, the encoder is represented by a stack of recurrent layers with LSTM cells and the decoder is the connectionist temporal classification (CTC) (Graves et al. (2006)). The CTC loss function computes the conditional probability by marginalizing all possible alignments and it assumes conditional independence between output predictions at different time steps given aligned inputs. An extra blank label, which can be interpreted as no label, is introduced to map $h$ and $y$ to the same length (i.e., an alignment or path). $a$ is obtained by inserting ($t'$ - $u$) blanks into $y$. A mapping $\mathcal{B} : a \to y$ is defined between $a$ and $y$, which can be done by removing all blanks and repeating letters in $a$.

$$P_{\text{CTC}}(y|x) = \sum_{a \in \mathcal{B}^{-1}(y)} P(a|h) \tag{1}$$

$$= \sum_{a \in \mathcal{B}^{-1}(y)} \prod_{t=1}^{t'} P(a_t|h_t) \tag{2}$$

$$P(a_t|h_t) = \text{softmax}(a_t, h_t) \tag{3}$$

---

[3]Training and data augmentation is performed using ResNet implementation in TensorPack

A.4.2 ATTENTION MODEL

Similar to the DS2 model, the attention model uses a stack of recurrent layers with GRU cells as the encoder. The decoder consists of an attention layer followed by a recurrent layer. The attention mechanism aligns the input sequence to the output sequence. The attention mechanism removes the conditional independence assumption in output sequence that the DS2 model makes. More model, attention mechanism, and loss function details can be found in Battenberg et al. (2017).

## B    USING VALIDATION ERROR

This section provides intuition about why learning curves that plot validation error against training data set size are relevant for deep learning.

Let $X$ be an input space and $Y$ be an output space. Let $Loss$ be a loss function.

Let $m$ be the number of training samples. Let $T_m = ((x_1, y_1), ..., (x_m, y_m)) \sim P$ be a sequence of training samples that is drawn independently from true distribution $P$, and $V \sim P$ be a validation set drawn independently from the same distribution $P$.

Let $M = DL\_ALG(S_m)$, where $DL\_ALG$ is a learning algorithm that yields a trained model $M_{DL\_ALG(S_m)} : X \rightarrow Y$, such as a deep neural network trained with stochastic gradient descent on training set $S_m$.

$$\mathcal{E}(M) := \mathbb{E}_{x,y \sim P}[Loss(M(x), y)] \tag{4}$$

Let $\mathcal{E}(M)$ be the expected risk, i.e. the expectation of evaluating the trained model $M_{DL\_ALG(S_m)}$ on the true distribution $P$.

$$\hat{\mathcal{E}}(M) := \frac{1}{m} \sum_{i=0}^{m} Loss(M(x_i), y_i), with\{(x_i, y_i)\}_{i=0}^{m} = S_m \tag{5}$$

Let $\hat{\mathcal{E}}(M)$ be the empirical risk.

In an empirical risk minimization setting, we are concerned with the generalization gap, i.e. $\mathcal{E}(M_{DL\_ALG(S_m)}) - \hat{\mathcal{E}}_{S_m}(M_{DL\_ALG(S_m)})$.

$$\hat{\mathcal{E}}_V(M) := \frac{1}{n} \sum_{i=0}^{n} Loss(M(x_i), y_i), with\{(x_i, y_i)\}_{i=0}^{n} = V \tag{6}$$

Let $\hat{\mathcal{E}}_V$ be the empirical risk on a held out validation set.

In practical deep learning settings, it is common to use $\hat{\mathcal{E}}_V$ to approximate $\mathcal{E}(M_{DL\_ALG(S_m)})$. This is a good approximation for sufficiently large validation sets if M is independent of V.

We use hyperparameter search including early stopping and model size selection to find models with the best validation error $\hat{\mathcal{E}}_V$ for each $S_m$. We assume that the limited number of evaluations performed does not introduce a significant dependence on V, which is aligned with best practices. In some cases, we use a completely different V for hyperparameter search.

In this setting, we rely on $\hat{\mathcal{E}}_V$ being a good approximation of $\mathcal{E}(M_{DL\_ALG(S_m)})$.

However, we still may need to contend with error due to optimization or error due to approximation.

Regarding these errors, it is well known that deep neural networks have sufficient capacity to overfit on many natural datasets, and reduce training error to zero, e.g. as studied in Zhang et al. (2017). Indeed for all of our studied tasks, we find that it is possible to train deep neural network models to reduce the training error below the validation error, resulting in overfitting.

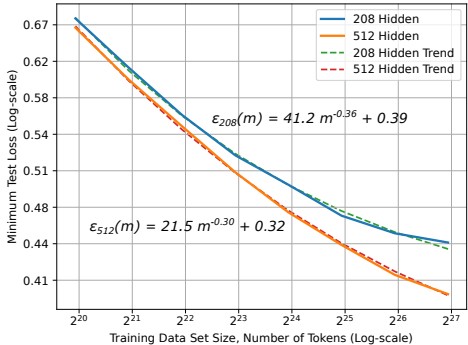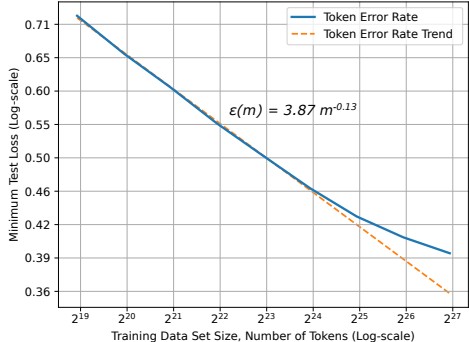

Figure 5: Neural machine translation learning curves. Left: the learning curves for separate models follow $\mathcal{E}(m) = \alpha m^{\beta_g} + \gamma$. Right: composite learning curve of best-fit model at each data set size.

So in this work we choose to characterize how $\hat{\mathcal{E}}_V$ scales with $m$, and how the model size scales with $m$. Such a characterization is of practical importance because it matches widely used DL methodology and allows interpretation and comparison of the relative difficulty of different tasks.

## C  OTHER TASKS

### C.1  NEURAL MACHINE TRANSLATION

One of our studied tasks is neural machine translation (NMT). Translation converts text input in one natural language to output text in another language. Relative to other tasks, NMT has low-dimensional input and output spaces, and can be trained with large labeled data sets. Our results show learning curve character similar to theoretical predictions, with a moderate exponent (i.e., $\beta_g \approx -0.128$).

To test NMT, we train a sequence-to-sequence model with global attention (Luong et al. (2015)) on the 2016 Conference on Machine Translation (WMT'16) German-to-English data set. We use a publicly available implementation of this architecture in OpenNMT (Klein et al. (2017)). The encoder contains two layers of bidirectional LSTMs, and the decoder contains the attention layer and stack of LSTM layers. To simplify training this model, we remove ensembling and data augmentation techniques (Sennrich et al. (2016b)).

To scale model sizes, we tie LSTM input and hidden state sizes together, and change them so that the total parameter count varies roughly linearly with a single scale factor parameter. We use Adam to optimize per-sequence cross-entropy loss and report the per-token classification error. We select models using the newstest2015 validation set, and we use the other newstest development sets from 2009 to 2013 for evaluation. Results presented here are with dropout rate of 0.2, though we tested without dropout and found similar learning curve exponents.

We clean and tokenize the data set using Moses (Koehn et al. (2007)) as described by Luong et al. (2017). We use the byte-pair encoding (BPE) method described by Sennrich et al. (2016a) to build a shared word-piece vocabulary between English and German. After preprocessing, the training set includes 4.5 million training sequences with roughly 130 million tokens in each language. We uniformly randomly shuffle the training data and sample training shards as described in Section 3.

Prior work predicts that as a model runs out of capacity on larger data sets, the error should plateau, resulting in a power-law + constant, $\mathcal{E}(m) \sim \alpha m^{\beta_g} + \gamma$, where $\gamma$ is the error when the model family has exhausted its capacity. Note the addition $\gamma$ term compared to the model used in other experiments.

Indeed, we find that learning curves for a single model family can be closely represented by a power-law + constant. We start by training fixed size models on each of the training shards. The left plot

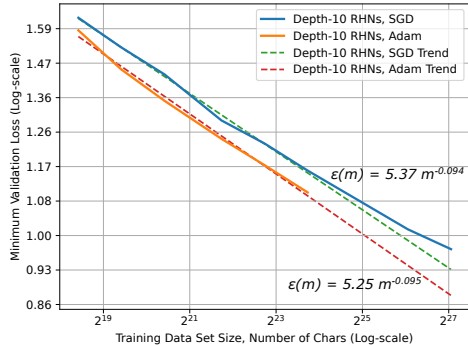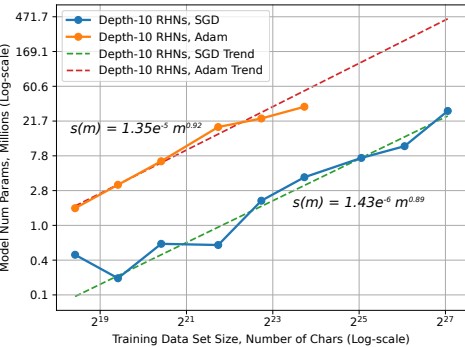

Figure 6: Learning curve (left) and model size (right) results and trends for character language models. Note the difference in absolute accuracy and model size, but same $\beta_g$ and $\beta_p$ between SGD and Adam optimizers.

in Figure 5 shows the learning curves for two different model sizes with 208 or 512 hidden nodes per LSTM layer (17M and 48M parameters, respectively). Learning curves with $\beta_g = -0.360$ and $-0.300$, respectively, fit the empirical results with less than 0.6% relative root mean square error.

The right plot in Figure 5 shows the composite learning curve for the best model at each shard size. Note the lack of a $\gamma$ term in this model. The best-fit results form a longer power-law region. We find that $\beta_g$ is even smaller than the single-model learning curves; if we project forward, $\beta_g$ would be approximately $-0.128$.

We also note that as training set sizes grow, optimization becomes more difficult and computational requirements limit the turn around time for experiments, so the empirical error tends away from the power-law trend. This divergence is common across domains, and we expect to need a more exhaustive hyperparameter search to find results closer to the existing power-law.

## C.2 CHARACTER LANGUAGE MODELS

To test character-level language modeling, we train RHNs of depth 10, which we found to achieve high accuracy on the Billion Word data set. We scale the number of layer weights to modulate the model size and find the best fit model for each training shard size. We use SGD, optimizing for per-predicted-character cross-entropy loss, which we report on the validation set. We also compare SGD against the Adam optimizer to test their effects. The input and output vocabulary includes all alphanumeric characters and common symbols for total size 98. We train the models on shards of 0.01% up to 4% of the Billion Word data set.

Results for character LMs appear substantially similar to word LMs. Figure 6 plots the validation and model size scaling results for character LMs. As with word LMs, validation error improves on a power-law as training data size increases, though the exponent is $\beta_g = -0.0936$ for the SGD optimizer and $\beta_g = -0.0954$ for the Adam optimizer. These power-law exponents are very similar despite the significant optimizer differences—Adam appears to just shift the learning curve down by $\sim 5\%$ relative.

Like word LMs, character LMs also learn significantly more slowly than predicted by theoretical results. Though word and character LMs have some major differences, their learning curve exponent differences indicate that character LMs are able to learn relationships between characters with successively fewer samples than word LMs are able to learn relationships between words.

Character LMs also show sublinear model size growth as data set size increases. Specifically, $\beta_p = 0.78$ for SGD optimized models and $\beta_p = 0.92$ for Adam optimized. Character LMs with the SGD optimizer see similar improvements from increased model size as word LMs, while the Adam optimized models see poorer scaling and require significantly more parameters ($\sim 8$–$11\times$). Still, their learning and model size curves appear predictable.

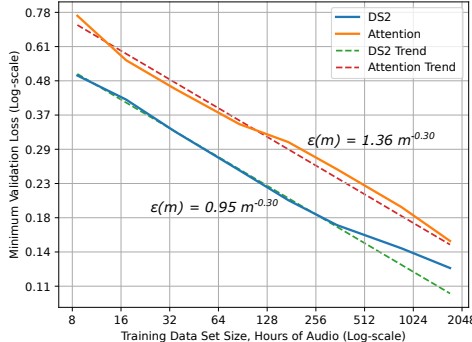 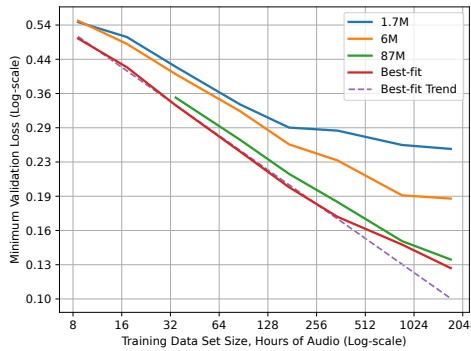

Figure 7: Learning curves for DS2 and attention speech models (left), and learning curves for various DS2 model sizes, 1.7M to 87M parameters (right). Note the similar $\beta_g$ despite of significant difference in training loss (CTC vs CE) and model architecture (CNN+RNN vs RNN+Attention).

## C.3 SPEECH RECOGNITION

Speech recognition techniques convert acoustic speech signals into text or commands. Speech recognition is used in diverse applications such as voice-powered machine controls and conversational user interfaces. Recent research has shifted from hand-engineered speech recognition pipelines over to end-to-end deep learning based methods that show promising results (Hannun et al. (2014); Chorowski et al. (2015); Amodei et al. (2016)). Speech input data is medium-dimensionality time-series data.

To test trends in speech recognition, we train two recent models: a variant of Deep Speech 2 (DS2) and an attention-based model. Our DS2 model (Amodei et al. (2016)) consists of two 2D convolution layers followed by four bidirectional LSTM recurrent layers. We use Adam to optimize connectionist temporal classification loss (CTC, Graves et al. (2006)). We compare DS2 against a hybrid attention model similar to those described by Battenberg et al. (2017). The model has an encoder comprised of three bidirectional LSTM layers with two intermediate max-pooling layers, and a hybrid attention decoder. We use Adam to optimize output sequence average cross-entropy loss. For both models, we remove regularization (weight decay and noise).

The inputs to these models are a sequence of log-spectrograms of power normalized audio clips, calculated on 20 ms windows. Outputs are the English alphabet along with the blank symbol. We do *not* include language models for output sequence beam search, and we report per-predicted-output character error rate on the validation set. We train on shards of labeled data set comprising 11,940 hours of speech containing 8 million utterances Amodei et al. (2016).

To vary the number of parameters in both the DS2 and attention models, we vary the number of weights in all LSTM layers, so that separate layers have the same number of weights. In the attention model, we also proportionally scale number of weights in the attention LSTM and decoder cells. For the DS2 model, model sizes range between 300K to 193M parameters, and for the attention based models, sizes range from 95K to 156M parameters.

Figure 7 (left) shows that both DS2 and attention based speech models experience the same power-law learning curve improvements. Although these models have significantly different encoders and decoders, they see the same relative improvements in character error rate as training set size increases with $\beta_g = -0.299 \pm 0.7\%$. Consistent with prior work (Bahdanau et al. (2016); Battenberg et al. (2017)), larger attention models trained on larger data sets tend to be easier to optimize than DS2 models, whose validation error tends away from the power-law trend on larger data sets.

