# OpenReview forum: "A Proposed Hierarchy of Deep Learning Tasks"
_ICLR.cc/2019/Conference_

### Official Review · AnonReviewer2 · 2018-10-28
**Interesting paper, but requires more work**

**Rating:** 4
**Confidence:** 2

**Review:**

The authors propose to measure the power-law exponent to sort natural language processing, speech and vision problems by the degree of their difficulty. The main idea is that, while in general model performance goes up for most tasks if more training data becomes available or for bigger model sizes, some tasks are more effective at leveraging more data. Those tasks are supposed to be easier on the proposed scale.

The main idea of the paper is to consider the bigger picture of deep learning research and to put different results on different tasks into an overall context. I think this is an exciting direction and I strongly encourage the authors to continue their research. However, the paper in its current state seems not quite ready to me. The write-up is repetitive at times, i.e., 'There is not data like more data.' appears 7 times in the paper. Also, some parts are very informal, e.g., the use of 'can't' instead of 'cannot'. Also, the sentence 'It would be nice if our particular proposal is adopted, but it is more important to us that the field agree on a satisfactory solution than that they adopt our particular proposal.', though probably correct, makes the reader wonder if the authors do not trust their proposal, and it would better be replaced by alternative suggestions or deleted. Also, the claim 'This may help explain why there is relatively more excitement about deep nets in speech and vision (vs. language modeling).' seems strange to me - deep nets are the most commonly used model type for language modeling at the moment.

Furthermore, I believe that drawing conclusions about tasks with the proposed approach is an over-simplification. The authors should probably talk about difficulties of datasets, since even for the same task, datasets can be of varying difficulty. Similarly, it would have been nice to see more discussion on what conclusions can be drawn from the obtained results; the authors say that they hope that 'such a hierarchy can serve as a guide to the data and computational requirements of open problems', but, unless I missed this, it is unclear from the paper how this should be done.

---

### Official Review · AnonReviewer1 · 2018-11-01
**ambitious goal and lack of approach**

**Rating:** 4
**Confidence:** 5

**Review:**

The paper provides empirical evidence that the generalization error scales inversely proportional to the log of number of training samples.

The motivation of the paper is well explained. A large amount of effort is put into experiments. The conclusion is consistent throughout.

It is a little unclear about the definition of s(m). From the definition at the end of Section 2, it is unclear what it means to fit the training data. It can mean reaching zero on the task loss (e.g., the zero-one loss) or reaching zero on the surrogate loss (e.g., the cross entropy). I assume models larger than a certain size should have no trouble fitting the training set, so I'm not sure if the curve, say in Figure 2., is really plotting the smallest model that can reach zero training error or something else.

Varying the size of the network is also tricky. Most papers, including this one, seem to be confined by the concept of layers. Increasing the number of filters and increasing the number of hidden units are actually two very structured operations. We seldom investigate cases to break the symmetry. For example, what if the number of hidden units is increased in one layer while the number is decreased in another layer? What if the number of hidden units is increased for the forward LSTM but not the backward? Once we break the symmetry, it becomes unclear whether the size of the network is really the right measure.

Suppose we agree on the measure of network size that the paper uses. It is nice to have a consistent theory about the network size and the generalization error. However, it does not provide any reason, or at least rule out any reason, as to why this is the case. For example, say if I have a newly proposed model, the paper does not tell me much about the potential curve I might get.

The paper spends most of the time discussing the relationship between the network size and the generalization error, but it does not have experiments supporting the hypothesis that harder problems are more difficult to fit or to generalize (in the paper's terminology, large beta_g and large beta_p). For example, a counter argument would be that the community hasn't found a good enough inductive bias for the tasks with large beta_g and beta_p. It is very hard to prove or disprove these statements from the results presented in the paper.

This paper also sends a dangerous message that image classification and speech recognition are inherently simpler than language modeling and machine translation. A counter argument for this might be that the speech and vision community has spent too much optimizing models on these popular data sets to the point that the models overfit to the data sets. Again these statements can be argued either way. It is hard to a scientific conclusion.

As a final note, here are the quotes from the first two paragraphs.

"In undergraduate classes on Algorithms, we are taught how to reduce one problem to another, so we can make claims about time and space complexity that generalize across a wide range of problems."

"It would be much easier to make sense of the deep learning literature if we could find ways to generalize more effectively across problems."

After reading the paper, I still cannot see the relationships among language modeling, machine translation, speech recognition, and image classification.

---

### Official Review · AnonReviewer3 · 2018-11-03
**Interesting approach for a relatively unexplored issue**

**Rating:** 6
**Confidence:** 3

**Review:**

The paper proposes an empirical solution to coming up with a hierarchy of deep learning tasks or in general machine learning tasks. They propose a two-way analysis where power-law relations are assumed between (a) validation loss and training set size, and (b) the number of parameters of the best model and training set size. The first power-law exponent, \beta_g, indicates how much can more training data be helpful for a given task and is used for ordering the hardness of problems. The second power-law exponent, \beta_p, indicates how effectively does the model use extra parameters with increasing training set (can also be thought of as how good the model is at compression). From experiments across a range of domains, the authors find that indeed on tasks where much of the progress has been made tend to be ones with smaller \beta_g (and \beta_p). It's arguable as to how comparable these power-law exponents are across domains because of differences in losses and other factors, but it's definitely a good heuristic to start working in this direction.

Clarifications needed:
(a) Why was full training data never used? The plots/analysis would have looked more complete if the whole training data was used, wondering why certain thresholds for data fraction were chosen.
(b) What exactly does dividing the training set into independent shards mean? Are training sets of different sizes created by random sampling without replacement from the whole training set?
(c) How exactly is the "Trend" line fitted? From the right sub-figure in Figure 1, it seems that fitting a straight line in only the power-law region makes sense. But that would require determining the start and end points of the power-law region. So some clarification on how exactly is this curve fitting done? For the record, I'm satisfied with the curve fitting done in the plots but just need the details.

Major issues:
(a) Very difficult to know in Figure 3 (right) what s(m)'s are associated with which curve, except the one for RHNs maybe.
(b) Section 4.1: In the discussion around Figure 2, I found some numbers a little off. Firstly, using the left plot, I would say that at even 8 images the model starts doing better than random instead of <25 that's stated in the text.  Secondly, the 99.9% classification error rate for top-5 is wrong, it's 99.5% for top-5 ("99.9% classification error rate for top-1 and top-5").
(c) Section 5: The authors use the phrase "low dimensional natural language data" which is quite debatable, to say the least. The number of possible sentences of K length with vocabulary |V| scale exponentially |V|^K where |V| is easily in 10K's most of the time. So to say that this is low dimensional is plain wrong. Just think about what is the (input, output) space of machine translation compared to image classification.

Typos/Suggestions:
(a) Section 3: "Depending the task" -> "Depending on the task"
(b) Section 4.3: "repeatably" -> "repeatability"
(c) Figure 4: Specify number of params in millions. The plot also seems oddly big compared to other plots. Also, proper case the axis labels, like other plots.
(d) Section 4.2 and 4.2.1 can be merged because the character LM experiments are not discussed in the main text or at least not clearly enough for me.  The values of \beta_g seem to include the results of character LM experiments. So either mention the character LM experiments in more detail or just point to results being in appendix.

---

### Meta-Review · Area_Chair1 · 2018-12-14

**Confidence:** 4
**Recommendation:** Reject

**Metareview:**

This paper attempts at ranking of tasks handled by deep learning methods based on learning curves.  A main premise of the paper is "fitting learning curves to a power law, and then sorting tasks by empirical estimates of exponents".   The idea of the paper is quite interesting.

However, the paper makes some bold claims which are a bit distant from the empirical study it conducts.  It is hard to line up the order in Table 2 with the Chomsky hierarchy.  Also, for various tasks, various different deep models are used (ResNets for image classification, LSTMs for LM, and so on).  I was not convinced that the beta parameter is model-agnostic.

Similar concerns are expressed by the reviewers, and they agree that the paper should address the criticism that they express in their feedback.